# Efficacy of Cognitive Behavior Therapy in Reducing Depression among Patients with Coronary Heart Disease: An Updated Systematic Review and Meta-Analysis of RCTs

**DOI:** 10.3390/healthcare11070943

**Published:** 2023-03-24

**Authors:** Aan Nuraeni, Suryani Suryani, Yanny Trisyani, Yulia Sofiatin

**Affiliations:** 1Doctoral Program, Faculty of Medicine, Universitas Padjadjaran, Bandung 45363, Indonesia; 2Department of Critical Care and Emergency Nursing, Faculty of Nursing, Universitas Padjadjaran, Bandung 45363, Indonesia; yanny.trisyani@unpad.ac.id; 3Department of Mental Health Nursing, Faculty of Nursing, Universitas Padjadjaran, Bandung 45363, Indonesia; suryani@unpad.ac.id; 4Department of Epidemiology, Faculty of Medicine, Universitas Padjadjaran, Bandung 45363, Indonesia; y.sofiatin@unpad.ac.id

**Keywords:** cognitive behavior therapy, coronary heart disease, systematic review, meta-analysis, depression

## Abstract

Purpose: The aim of this review is to identify the efficacy of cognitive behavior therapy (CBT) and the characteristics of CBT therapy that effectively improve depression among patients with coronary heart disease (CHD). Methods: Studies that assessed CBT efficacy in decreasing depression among CHD patients with randomized controlled trials (RCTs) were searched through PsycINFO, PubMed, CINAHL, Academic Search Complete, Scopus, and Google Scholar. Two reviewers independently screened and critically appraised them using the Cochrane risk-of-bias tool. The fixed- and random-effect models were applied to pool standardized mean differences. Results: Fourteen RCTs were included in the quantitative analysis. Depression was significantly lower in the CBT group (SMD −0.37; 95% CI: −0.44 to −0.31; *p* < 0.00001; I^2^ = 46%). Depression in the CBT group was significantly lower in the short-term follow-up (SMD −0.46; 95% CI: −0.69 to −0.23; *p* < 0.0001; I^2^ = 52%). Moreover, the subsequent therapy approaches were effective in reducing depression, including face-to-face and remote CBT, CBT alone or combination therapy (individual or mixed with a group), and frequent meetings. Conclusions: CBT therapy effectively reduces depression, particularly in short-term follow-up. The application of CBT therapy in CHD patients should consider these findings to increase the efficacy and efficiency of therapy. Future research is needed to address generalizability.

## 1. Introduction

Cardiovascular diseases (CVDs), including coronary heart disease (CHD), are the most common cause of death globally and decrease individuals’ quality of life [1]. Approximately 80% of deaths caused by CVD occur in low- and middle-income countries due to increased CVD risk factors [1]. These risk factors also escalate the prevalence of CHD in the future [2]. CHD has a high recurrence rate and can be exacerbated by psychological problems such as anxiety, depression, helplessness, and stress [3,4,5,6,7].

Patients with CHD often experience psychological problems, particularly depression. Several studies have revealed a high prevalence of depression followed by increased mortality and morbidity rates. Several studies on quality of life have indicated that depression is a significant predictor of poor quality of life in patients with CHD [8,9,10]. Moreover, according to Lichtman et al. and Vaccarino et al., as many as 15–30% of patients experience depression [11,12]. Depression is also associated with the severity of functional impairment, low adherence to therapy, and low participation in cardiac rehabilitation [12], as well as increased morbidity and risk of death [11,12]. These conditions emphasize a pronounced need for depression recognition and improvement in depression management for patients with CHD [11,13].

Treatment of depression in CHD patients classified in various studies generally consists of psychosocial therapy [14,15,16], pharmacological therapy [17], exercises [18,19], nutrition [20], and spiritual therapy [21,22]. Of these many therapies, psychosocial therapy is the most frequently given therapy to reduce depression in patients with cardiac problems. The former systematic reviews and meta-analyses identified the efficacy of psychosocial interventions for depression and psychosocial outcomes in CHD patients [23,24]. These studies generally identified psychosocial therapy and its effects on psychological problems. The results highlighted the potential efficacy of psychosocial interventions in improving patients’ psychological issues. Cognitive behavioral therapy (CBT) is a psychosocial intervention that is widely used to reduce depression in CHD patients.

The effect of CBT on depression in CHD patients was identified in a systematic review and meta-analysis. The results of this review indicated that CBT has the potential to be effective in reducing depression in CHD patients [25]. However, several limitations were identified in this meta-analysis, including the small number of studies, which were limited to two countries in Europe, as well as the USA and Australia. Hence, adding the latest studies might improve the generalizability aspect. In addition, the heterogeneity of this review was still quite high (I^2^ = 59%); thus, the analysis used to identify the effect of CBT on depression used a random-effect model. This approach has more weaknesses compared to the fixed-effect model, whereby the uncertainty in the random-effect model includes uncertainty in the sampling or estimation aspects coupled with uncertainty resulting from variations between studies. In contrast to the fixed-effect model, the source of uncertainty only occurs because the uncertainty is only in the sampling aspect [26]. This shows that there is still an opportunity to add new studies to reduce the possibility of errors in variations from the results of previous meta-analyses. In addition, the results of the prior review did not identify the effectiveness of internet-based CBT in reducing depression among CHD patients. However, therapy delivery using the internet can provide benefits for broader and more affordable interventions. Accordingly, data related to the effectiveness of therapy using the internet must be examined.

Furthermore, the variety of therapy among studies, such as single or combination treatment, various intervention frequencies, and different follow-up periods, caused high heterogeneity in the prior meta-analysis; consequently, conclusions based on the analysis results for several outcomes need to be considered with caution [25]. Therefore, we aimed to perform an updated systematic review and meta-analysis with the primary outcome of identifying the efficacy of CBT in depression among CHD patients. The secondary outcome was to identify the characteristics of CBT therapy that effectively reduce depression in CHD patients.

## 2. Materials and Methods

### 2.1. Study Design

This study was a systematic review conducted following the Preferred Reporting Items for Systematic Reviews and Meta-Analyses guidelines [27]. The study protocol was not published or registered.

### 2.2. Eligibility Criteria

The research questions and eligibility criteria for this study used the PICOT (population/intervention/comparator(s)/outcomes/type of study) approach. Our criteria for including studies in this analysis were randomized controlled trials (RCTs) published until October 2022 that identified the effect of cognitive behavioral therapy (CBT) on depression as an outcome in patients with CHD. We excluded protocol, non-English language studies, and non-accessible full-text publications. Furthermore, there was no restriction on the year of publication. The PICOT approach applied in the review is described below.

#### 2.2.1. Types of Participants

Participants in the included studies were patients with CHD. They could be patients with stable angina or patients with acute coronary syndrome (ACS) (unstable angina, ST-elevation myocardial infarction (STEMI), or non-ST-segment elevation myocardial infarction (NSTEMI)), regardless of whether they had undergone reperfusion therapy such as percutaneous coronary intervention (PCI) or coronary artery bypass graft (CABG). Patients had a depression score or identifiable depressive symptoms based on a reliable and valid depression measurement tool.

#### 2.2.2. Types of Interventions

The intervention used was CBT or therapy based on CBT principles, which was carried out as the primary therapy or in combination with other therapies. CBT emphasizes efforts to change mindsets and behavior. This therapy focuses on helping individuals learn to be their therapist. CBT is provided through in-session exercises and “homework” outside sessions to help patients/clients develop their coping skills, where they can learn to change their problematic thoughts, emotions, and behaviors [28].

#### 2.2.3. Types of Comparators

The comparator was a waitlist intervention, usual care, or other interventions such as medication and other psychosocial therapy. Furthermore, in the studies analyzed, there were four types of comparators: usual care [29,30,31,32,33,34,35,36,37,38,39,40], waitlist intervention [41], brief intervention [42], and clinical management [43].

#### 2.2.4. Types of Outcome Measures

The primary outcome of this review was the efficacy of CBT therapy in reducing depression or depressive symptoms among CHD patients. Moreover, the secondary outcome was identifying the efficacy of CBT in reducing depression or depressive symptoms based on follow-up time: at 1–3 months of follow-up (short-term effect), at 6 months of follow-up (medium-term effect), and at more than 9 months of follow-up (long-term effect). In addition, this study identified the efficacy of CBT delivered remotely via telephone and the internet, the efficacy of CBT given individually, in groups, or mixed, and the effectiveness of CBT based on the frequency of meeting in reducing depression in CHD patients.

#### 2.2.5. Types of Studies

Our criteria for including studies in this analysis were RCTs published in English that identified the effect of CBT on depression as an outcome in patients with CHD; we excluded protocol studies. Furthermore, there was no publication year restriction.

### 2.3. Data Collection and Analysis

#### 2.3.1. Search Strategy

Two authors independently performed a structured literature search (A.N. and S.S.) using five databases: CINAHL (accessed on 30 October 2022), PubMed (accessed on 30 October 2022), Scopus (accessed on 30 October 2022), Academic Search Complete (accessed on 30 October 2022), and PsycInfo (accessed on 31 October 2022), as well as the search engine Google Scholar (accessed on 31 October 2022), with the following keywords: coronary heart disease, cardiac, heart, myocardial, psychosocial intervention, psychotherapy, CBT, cognitive behavior therapy, depressive, depression, experimental, and randomized trial. We also used the Boolean operators “OR” and “AND” in the literature search, generating the following search string: (coronary heart disease OR cardiac OR heart OR myocardial) AND (psychosocial intervention OR psychotherapy OR CBT OR cognitive behavior therapy) AND (depressive OR depression)) AND (experimental OR randomized trial) (Appendix A shows the search strategy used in each database). We also involved studies based on references from articles included in the analysis.

#### 2.3.2. Study Selection

Two independent reviewers (A.N. and S.S.) selected the studies. We first checked the articles included in the initial search for duplication using the Mendeley reference manager. We then checked the titles, abstracts, and full texts for the relevance regarding the study topic and the inclusion and exclusion criteria. Lastly, we checked each full text against the Joanna Briggs Institute (JBI) critical appraisal checklist for RCTs. Specifically, we calculated the critical appraisal scores as the number of “yes” responses divided by the total number of “unclear”, “no”, and “yes” responses, excluding any “no information” responses [44]. Following our appraisals, we eliminated any study with a score of <70%. Furthermore, a third author (Y.T.) provided a decision if there was a discrepancy in the selection results. Notably, we experienced no disagreements regarding paper eligibility.

#### 2.3.3. Data Extraction

The data were extracted using a data extraction table by one reviewer (A.N.) and checked by another reviewer (Y.S.). The extraction data compiled the information related to each study’s characteristics: study design, sample size, participants, comparator, interventions, and outcomes. The interventions consisted of the frequency of CBT, therapy approach (individual, group, or mixed), therapy delivery (face-to-face, telephone-based, internet, or mixed), and CBT mono- or combination therapy. The primary and secondary outcomes were identified and extracted using standard mean differences and standard deviations according to the baseline and/or follow-up.

#### 2.3.4. Assessment of Risk of Bias in Included Studies

Two reviewers (A.N. and Y.T.) independently assessed the risk of bias (RoB) for RCTs in included studies using the Cochrane risk-of-bias (RoB) tool, consisting of five domains: randomization process, deviations from intended intervention, missing outcome data, measurement of the outcome, and selection of the reported result [45]. RoB was assigned as “high”, “low” or “unclear”, or “no information” for each domain. Discrepancies in the assessment results were then discussed, and a third reviewer (S.S.) determined the decision.

#### 2.3.5. Data Synthesis

The synthesized data used changes in the standard mean difference and standard deviation with a 95% confidence interval (CI) and the *p*-values in pre–post intervention in the treatment and control groups extracted from included studies using Review Manager ver. 5.4 (The Nordic Cochrane Center, The Cochrane Collaboration, Copenhagen, Denmark). Then, we interpreted the pooled effects using the inverse variance method. CBT’s efficacy or effect estimation in reducing depression among patients with CHD was determined by the standard mean difference with a 95% CI and its respective *p*-value in pre–post intervention in the CBT and control groups. Heterogeneity was evaluated by observing the overlap of confidence intervals on the forest plots, and the I^2^ value, with meta-analysis, was considered inappropriate if there was substantial heterogeneity (I^2^ ≥ 75%) [46]. In addition, if the I^2^ value was <50%, the analysis used a fixed-effect model; if the I^2^ value was >50%, the analysis used a random-effect model.

We minimized heterogeneity in several ways. When studies had multiple trial arms, the CBT intervention was compared to usual care. If outcomes were identified at various timepoints, the initial outcome measure was taken in the analysis. Moreover, if studies conveyed multiple depression scores from different scales, the most common scale used by other eligible studies was elected [25]. We also used the Jamovi version 2.3.21 software when the heterogeneity was high to identify any outlier studies.

#### 2.3.6. Publication Bias

A funnel and forest plot and Egger’s test using Jamovi version 2.3.21 software were performed to assess publication bias. The results demonstrated no indication of outliers if neither the rank correlation nor the regression test indicated any funnel plot asymmetry (*p* > 0.05) [46].

#### 2.3.7. Subgroup Analysis

A subgroup analysis was performed to identify CBT’s effectiveness based on various delivery approaches in reducing depression in patients with CHD, e.g., the follow-up depression scores, frequency of CBT, therapy approach (individual, group, or mixed), therapy delivery (face-to-face, telephone-based, internet, or mixed), and therapy type (single or combination CBT). Analysis was also carried out to see the effect of CBT on specific populations of patients with CHD, such as post-CABG patients.

#### 2.3.8. Sensitivity Analysis

The sensitivity analysis was identified by comparing the analysis results while excluding studies with low risk of bias vs. high risk of bias.

## 3. Results

### 3.1. Study Selection

We identified a total of 1578 articles in the initial search: 52 from PsycINFO, 338 from Academic Search Complete, 272 from CINAHL, 68 from Scopus, and 748 from PubMed. We also identified 64,000 publications from Google Scholar, of which we selected the first 100 publications, which were sorted on the basis of their level of relevance. We excluded 35 duplicate articles and another 1517 studies because the articles’ title and information in the abstract were irrelevant since they were not intervention studies, the respondents were non-cardiac patients, or the research method was a review. According to our inclusion and exclusion criteria, 15 articles remained; through a manual search, we found one additional eligible article. Thus, 16 articles were included in the qualitative analysis, while only 14 articles were included in the quantitative analysis because we failed to find the mean difference score from one article [38], and there was one article with publication bias [32]. Figure 1 displays the study selection procedure used through a PRISMA flow diagram.

### 3.2. Characteristics of the Included Studies

All the study designs included in the analysis were randomized controlled trials (RCTs). Of the 16 articles included in the qualitative analysis, 5 were conducted in the United States, along with 2 in Australia, 3 in Sweden, and 1 each in New Zealand, Portugal, Iran, Scotland (UK), and Italy. Characteristics of the included studies are shown in Appendix A.

#### 3.2.1. Participants

Participants were patients with CHD, including acute coronary syndrome (ACS), myocardial infarction (MI), angina, and unstable angina. Most of the participants were patients who had undergone treatment of recent acute myocardial infarction (AMI), post-PCI or CABG treatment [35,37,41], or both [38,42]. There were also CHD patients undergoing cardiac rehabilitation [36,47]. All participants were identified as having depression or depressive symptoms, the assessment of which was based on validated instruments for assessing depression.

#### 3.2.2. Intervention

The interventions used in studies were CBT alone or combined with other therapies such as wellbeing therapy [43], antidepressants, and education related to CHD management [34,38,39,40], as well as cardiac rehabilitation [34,36,47].

Twelve (75%) of the sixteen studies used CBT as primary treatment, although, in several studies, it was stated that other therapies were given [30,31,33,34,35,36,37,40,41]. Eleven studies mentioned that psychologists carried out therapy, while additional therapists were also involved: cardiologists, nurses, and social workers. The administration of therapy sessions in each study was different. We divided them according to the frequency of therapy. A low frequency of meetings constituted 3–6 sessions [34,37,42], a medium frequency constituted 8–12 sessions [30,31,33,35,36,41], and a high frequency constituted 13–24 sessions [32,38,39,40]. The delivery of therapy varied, but most meetings given face-to-face; this mode of delivery was reported in eleven studies, whereas three therapies used the phone [30,33,36], and four were given via the internet [38,39,40,42]. In addition, individual therapy administrations were reported by nine studies, intervention delivery in a group was reported by four studies [31,40,42,43], and a combination of the two (i.e., group sessions followed by individual sessions) was reported by three studies [32,34,36].

#### 3.2.3. Comparator

In the studies analyzed, there were four types of comparators: usual care [30,31,32,33,34,35,36,37,38,39,40], waitlist intervention [41], brief intervention [42], and clinical management [43]. Most of the identified comparators mentioned in the studies were usual care, where the intervention given was in accordance with the treatment program received by the patient at the hospital or cardiac rehabilitation program such as therapy to reduce risk factors and promote a healthy lifestyle. Another comparator was the provision of brief interventions in the form of giving feedback from the baseline assessment, where the control group did not receive the subsequent CBT treatment [42].

### 3.3. Risk of Bias within Studies

The bias assessment results showed a low risk of bias in measuring the outcome data. Even though outcome assessors were aware of the intervention received, it is unlikely that this affected the outcome assessment. The knowledge provided with the measured outcome may have differed because the outcome took into account their psychological condition. Most studies had a low risk of bias related to the randomization process. Only three studies showed a moderate risk of bias because they did not clearly explain how random allocation and concealment were carried out [31,34,37]. In addition, most studies also had a low risk of bias in the missing outcome data because the necessary data related to depression scores at baseline and follow-up were available. A total of six studies showed a high risk of bias, while three showed a moderate risk of bias in terms of deviation from the intended intervention because almost all participants were aware of the interventions given, and only a few studies implemented personnel blinding. In addition, four studies reported noncompliance with intervention [31,38,39,42]. Regarding selection bias, a moderate risk of bias may have occurred in nine studies because there was no protocol described [30,32,33,35,38,39]. The bias assessment results are presented in Figure 2 and Figure 3.

### 3.4. Sensitivity Analysis

Sensitivity was identified by comparing changes in the follow-up results while excluding all studies with a low risk of bias vs. a high risk of bias in the analysis. The analysis results of all studies did not change significantly when studies with a low risk of bias [30,31,33,48] were omitted (SMD −0.38; 95% CI: −0.50 to 0.26; *p* < 0.00001). Similarly, when studies with a high risk of bias were excluded, no significant changes were found in the primary outcome (SMD −0.36; 95% CI: −0.43 to −0.30, *p* < 0.00001). These results are shown in Appendix A.

### 3.5. Publication Bias

Funnel plots and Egger’s tests were used to identify publication bias. Egger’s test analysis results showed a *p*-value of <0.001, indicating publication bias. Moreover, from the forest plot using Jamovi software, publication bias was identified in the Fernandes study [32]. Thus, we excluded this study and conducted a reassessment; the results of Egger’s test showed *p* > 0.01, indicating no publication bias (Table 1), In accordance with the funnel plot in Figure 4 which depicts the distribution of the studies based on their standardized mean difference and standard deviation. Lastly, the number of studies included in the quantitative analysis was 14 articles. The forest and funnel plots results of publication bias with the outlier study are shown in Appendix A.

### 3.6. Efficacy of CBT in Reducing Depression among Patients with CHD

The results of the meta-analysis of 14 studies with a total of 3795 participants (CBT n = 1878, control n = 1917) showed that the pooled depression score in patients with CHD after follow-up in the CBT group was significantly lower than in the control group (SMD −0.37; 95% CI: −0.44 to −0.31; Z = 11.29; *p* < 0.00001; I^2^ = 46%) (Figure 5). This result demonstrates that CBT has good efficacy in reducing depression in CHD patients.

### 3.7. Subgroup Analysis

In the short-term follow-up (≤3 months after intervention) (Appendix A), it was identified that CBT had a superior effect compared to the control group (SMD −0.46; 95% CI: −0.69 to −0.23; Z = 3.95; *p* < 0.0001; I^2^ = 52%). Similarly, in the medium-term follow-up (6 months after intervention) (Appendix A), CBT was still more effective in reducing depression than the control group (SMD −0.40; 95% CI: −0.76 to −0.04; Z = 2.19; *p* = 0.03; I^2^ = 89%). However, the result had a substantial heterogeneity of more than 75%; hence, conclusions should be drawn with caution.

The results of long-term follow-up (≥12 months after intervention) (Appendix A) showed no significant effect on depression scores in the CBT group compared to the control group (SMD −1.02; 95% CI: −2.05 to 0.01; Z = 1.93; *p* = 0.05; I^2^ = 96%). Nevertheless, because the heterogeneity was very high, pooling was considered inappropriate. An analysis based on the number of remissions of depression in the long-term follow-up based on two studies [35,42] showed that the remission results were greater in the intervention group. However, the *p*-value was insignificant with substantial heterogeneity (OR 2.45; 95% CI: 0.41 to 14.48; Z = 0.99; *p* = 0.32; I^2^ = 80%).

According to the delivery approach, face-to-face CBT therapy (Appendix A) had a significant effect on reducing depression scores at follow-up compared to the control group (SMD −0.36; 95% CI: −0.43 to −0.29; Z = 9.75; *p* < 0.00001; I^2^ = 36%). Likewise, the remote therapy delivery mode (by phone or internet) (Appendix A) was more effective in reducing depression than the control group (SMD −0.32; 95% CI: −0.47 to −0.17; Z = 4.15; *p* < 0.0001; I^2^ = 49%). Yet, according to the effect size and SMD, face-to-face therapy still had a better effect than remote therapy. Subgroup analysis of remote therapy using the internet (Appendix A) also showed greater effectiveness compared to the control group (SMD −0.41; 95% CI: −0.60 to −0.22; Z= 4.27; *p* < 0.0001; I^2^ = 0%).

Studies were divided according to the frequency of meetings into three categories: low, medium, and high frequency. Low-frequency CBT (3–6 sessions) (Appendix A) was not effective in reducing depression (SMD −0.39; 95% CI: − 0.93 to 0.15; Z = 1.43; *p* = 0.15; I^2^ = 71%), whereas moderate-frequency CBT (8–12 sessions) (Appendix A) appeared to be effective in reducing depression (SMD −0.35; 95% CI: −0.65 to −0.05; Z = 2.29; *p* = 0.02; I^2^ = 65%), as did high-frequency CBT (>13 sessions) (Appendix A) (SMD −0.36; 95% CI: −0.43 to −0.29; Z = 9.62; *p* < 0.00001; I^2^ = 0%).

According to the therapy approach, it was identified that individual (SMD −0.89; 95% CI: −1.43 to − 0.35; Z = 2.35; *p* = 0.02), I^2^ = 94%) (Appendix A) and mixed therapy (SMD −0.36; 95% CI: −0.43 to −0.28; Z = 9.04; (*p* < 0.00001; I^2^ = 0%) (Appendix A) had a significant effect on reducing depression. However, individual treatment approaches showed substantial heterogeneity (I^2^ > 75%), indicating variation in the population that needs to be specified. We excluded two studies with the CABG patient population [35,37], and the results showed significant efficacy with lower heterogeneity (SMD −0.35; 95% CI: −0.53 to −0.16; Z = 3.65; *p* = 0.0003; I^2^ = 39%) (Appendix A). In contrast, therapy using a group approach (Appendix A) showed insignificant effectiveness (SMD −0.07; 95% CI: −0.36 to 0.22; Z = 0.49; *p* = 0.63; I^2^ = 0%).

CBT therapy alone (SMD −0.40; 95% CI: −0.63 to −0.17; Z = 3.43; *p* = 0.0006; I^2^ = 61%) (Appendix A) and in combination with other therapies (SMD −0.36; 95% CI: −0.44 to −0.29; Z = 9.73; *p* < 0.00001, I^2^ = 0%) (Appendix A) showed significant effectiveness in reducing depression. In addition, the effectiveness of therapy indicated a higher standard mean difference in post-CABG patients (SMD −0.84; 95% CI: −1.11 to −0.57; Z = 6.11; *p* < 0.00001; I^2^ = 0%) (Appendix A) versus the CHD patient population on various treatments (post-PCI, CABG cardiac rehabilitation) (SMD −0.34; 95% CI: −0.41 to −0.28; Z = 10.13; *p* < 0.00001; I^2^ = 10%) (Appendix A).

## 4. Discussion

### 4.1. Principal Results

The primary purpose of this review was to provide an update of the efficacy of CBT in improving depression, and the secondary outcome was to analyze the characteristics of CBT that can effectively improve depression in patients with CHD. Fourteen studies included in the quantitative analysis in this study. In contrast to previous research, 10 of 12 studies discussed CBT’s effects on depression in patients with CHD [25]. Updates from this study were identified from four additional studies [30,33,38,39] in the period 2015 to 2022, whereas the former meta-analysis analyzed studies published from 2003 to 2014. The studies analyzed in this review came from four continents consisting of eight countries (the United States, Australia, Germany, Scotland, Sweden, Italy, New Zealand, and Iran), while the previous study only analyzed studies originating from three continents with four countries (Scotland, Germany, the USA, and Australia). The additional results in this review increase the external validity of the generalization aspect and increase the number of samples from different populations and countries, providing a more comprehensive.

The analyzed studies found that depression scores decreased in all participants over time. This condition may occur along with improving physical and psychosocial conditions following CHD. Patients with early-stage CHD can experience a high degree of helplessness after an acute attack, which is significantly associated with depression [49,50]. Notably, the helplessness gradually decreases after the patient’s condition improves, mainly through cardiac rehabilitation [51]. However, according to the results of this meta-analysis, it was identified that the CBT group had better efficacy in reducing depression than the control group. This review reinforces the results of previous meta-analyses [25]. It provides the latest updates regarding the heterogeneity of studies, the type of analysis used, and the characteristics of effective therapy in CBT, primarily highlighting the effect of internet-based CBT on reducing depression in CHD patients.

The results of this study indicated a slightly higher SMD with lower heterogeneity (−0.37; 95% CI: −0.44 to −0.31; *p* < 0.00001; I^2^ = 46%), compared to the previous study (SMD −0.35; 95% CI: −0.52 to −0.17; *p* < 0.001; I^2^ = 59%) [25]. Thus, this study used a fixed-effect model, in contrast to previous meta-analyses that used a random-effect model. In the fixed-effect model, all studies are assumed to have the same value, which yields a better estimation [52].

According to follow-up time, the effectiveness in reducing depression was significantly better at short-term and medium-term follow-up compared to long-term follow-up. Thus, CBT appears to have a rapid effect in reducing depression; however, after a more extended period, the effect is similar to the control group. Almost all studies showed an improvement in depression over time. This finding may be because depression with mild symptoms resolves spontaneously [48], whereas the studies analyzed provided less discussion on depression levels, such as major depression. Thus, a comparison of the effect of depression on the level of depression at each stage of follow-up could not be performed. In addition, interventions carried out in the control group that received standard therapy, such as cardiac rehabilitation, may also have had an emotional or psychological impact on the patient [30]. Future research needs to further explore the effects of CBT in patients with more severe depressive problems, such as major depressive disorder (MDD), to identify the effect of CBT on improving depression in CHD patients at short-, medium-, and long-term follow-up.

The previous meta-analysis did not identify the effectiveness of providing remote-CBT using the internet. However, in this study, it was shown that providing therapy through this remote approach effectively reduces depression in patients with CHD. Nonetheless, the identification of the effectiveness of CBT using this internet platform only emerged from two studies in the same country, Sweden, with a small number of participants; hence, external validity is weak. Thus, other studies need to be carried out to improve the generalizability aspect. Furthermore, CBT can also be provided remotely via telephone; according to the overall analysis (internet and phone), this remote CBT approach had a better effect compared to the control group (SMD −0.32; 95% CI: −0.47 to −0.17; Z = 4.15; *p* < 0.0001; I^2^ = 49%). The provision of remote CBT therapy can undoubtedly provide benefits because it can increase time and cost efficiency, as well as expand the range of therapy [25,38].

Face-to-face CBT therapy was the only therapeutic approach shown to have a better effect in previous studies, compared to telephone and mixed therapy (face-to-face and phone) [25]. In this study, different results were identified. Both therapy modes, face-to-face and telephone-based, had the same significant effect on reducing depression. In addition, face-to-face therapy in this study had a lower level of homogeneity (I^2^ = 36%) compared to previous studies (I^2^ = 60%), which allowed the analysis to be carried out using a fixed-effect model, providing a more accurate estimate [52]. The results indicated that CBT therapy, whether face-to-face or remote, is equally effective in reducing depression in CHD patients.

Effective results in reducing depression were found at moderate (8–12 sessions) and high (>13 sessions) frequencies. However, different outcomes were identified with fewer sessions. These results indicate that an increased frequency of therapy can have a better effect. More exercises can give participants a better understanding and skills for overcoming problems. Thus, the frequency of therapy is essential to be considered in treatment planning because it can have an impact on the efficiency and effectiveness of therapy [53]. These findings may provide an update regarding effectiveness related to the frequency/duration of courses, whereby prior studies found that short and long courses of therapy did not provide superior effects compared to the control group [25].

Another characteristic analyzed in this study was the provision approach of therapy. In the literature, CBT was administered using individual and mixed approaches, with both effectively reducing depression. On the other hand, the group approach did not provide significant effectiveness. However, despite the effectiveness of the results, analysis of the individual approach showed substantial heterogeneity with I^2^ > 75%. This very high heterogeneity was likely due to differences in the participant population across studies because, after we excluded studies with CABG patients, the heterogeneity rate decreased sharply to 39%. Another meta-analysis corroborates this result that individual CBT was more effective for individuals experiencing depression accompanied by physical problems [54], as experienced by patients with CHD.

Post-CABG patients have higher physical and emotional vulnerability in the early phases after surgery. This indication can be perceived from the lower quality of life in the early post-CABG stage compared to patients who have undergone PCI [55]. Exciting results were identified in this study regarding the population of participants with CABG. Heterogeneity increased when studies with CABG participants were included in the analysis. This condition may indicate differences in the characteristics of the participant population. Hence, it is crucial to conduct subgroup analysis for this population. In this study, there were two studies from the US involving patients with CABG; the results revealed CBT’s effectiveness in reducing depression in CABG patients. However, these studies are limited, with low generalizability. Therefore, future studies need to be performed in this patient population.

CBT alone and CBT combined with other therapies both showed better results compared to the control group. Nevertheless, a higher effect with less heterogeneity was seen in CBT combination therapy, including antidepressants [29,32], other psychotherapy [56], and wellbeing therapy [43]. Combination therapy showed more effective results; this is expected since CBT is known to be more effective in treating mild to moderate depression but insufficient for treating severe depression. According to Chand, patients with major depression require a combination of drug therapy and psychotherapy [48]. These outcomes were reinforced by Davidson et al., whose patients preferred choosing their therapy; if their chosen therapy (CBT or pharmacotherapy) did not have positive effects on their depressive symptoms within 6–8 weeks, care was taken to add another form of therapy, and the authors found that this approach was effective in reducing depression in patients with CHD [56].

In addition to the effectiveness of CBT and its characteristics in reducing depression in CHD patients, this study recognized several risk factors of reducing the efficacy of therapy. A decrease in efficacy was identified from two studies that revealed unsatisfactory results [36,42], which comprised interventions with a short duration of six sessions/weeks [42] and twelve sessions but with a short total intervention time of only 2.5 h [36]. In addition, one of the two studies used group intervention [42]. These two characteristics, i.e., a short intervention and a group approach, were shown to have no significant effect on reducing depression in this meta-analysis. This result is also reinforced by the research of Christensen et al., who concluded that extended CBT was more effective in reducing depression than brief CBT [57]. Likewise, some studies showed that individual therapy was more effective than group therapy [58]. However, other studies indicated similar effectiveness [59]. In summary, these two factors can influence the efficacy of CBT in reducing depression, although it is possible that additional factors remain unidentified. However, it is recommended to cautiously consider brief interventions and a group approach in CBT to reduce depression in this patient population.

### 4.2. Limitations

This study did not elucidate the effectiveness of CBT in long-term follow-up compared to the control group since both the control and the intervention groups had insignificant differences in scores. Further identification is needed regarding the factors influencing whether improvement occurs over time in mild depression, and whether improvement occurs spontaneously, with the influence of other therapies, or according to the severity of depression. In addition, the effectiveness of medium- and long-term follow-ups revealed high heterogeneity, indicating insufficient pooled data.

The weaknesses in the included studies might have arisen from low internal validity in some aspects. Some participants were aware of the intervention, which could have increased the risk of outcome assessment bias on the results. In addition, several studies showed low participant compliance with interventions, especially in internet-based CBT, which could have raised the risk of increased deviation from the intended intervention. Future research needs to improve these weaknesses by applying participant blinding and increasing participant compliance.

### 4.3. Implications for Clinical Practice

CBT is recommended for depressed patients with CHD to reduce depression, especially in the short term. In addition, several factors related to the therapeutic approach need to be considered for effective results, including the frequency of therapy (>8 sessions), therapy approach (individual or a combination of individual and group sessions), treatment type (alone or in combination with other treatments), and mode of therapy (face-to-face or remotely).

## 5. Conclusions

The findings of this systematic review and meta-analysis corroborate the results of previous meta-analyses where CBT therapy relieved depression or reduced depressive symptoms, especially in short-term follow-ups in CHD patients. In addition, this updated meta-analysis identified that internet-based CBT therapy also provides a significant therapeutic effect in reducing depression. Specific characteristics of the approach mediate the effectiveness of treatment, such as the frequency of meetings (>8 sessions), therapy approached (individual or a combination of individual and group sessions), treatment type (alone or in combination with other treatments, especially those related to lifestyle improvements, physical exercise, and antidepressants), and therapy mode (face-to-face or remotely via telephone or internet). Although several results were identified from this meta-analysis, not enough data have been collected related to medium- and long-term CBT effects in patients with CHD who experience more severe depression. Moreover, additional studies are needed on internet-based CBT. Therefore, future studies are needed to improve the generalizability and efficacy associated with these CBT effects.

## Figures and Tables

**Figure 1 healthcare-11-00943-f001:**
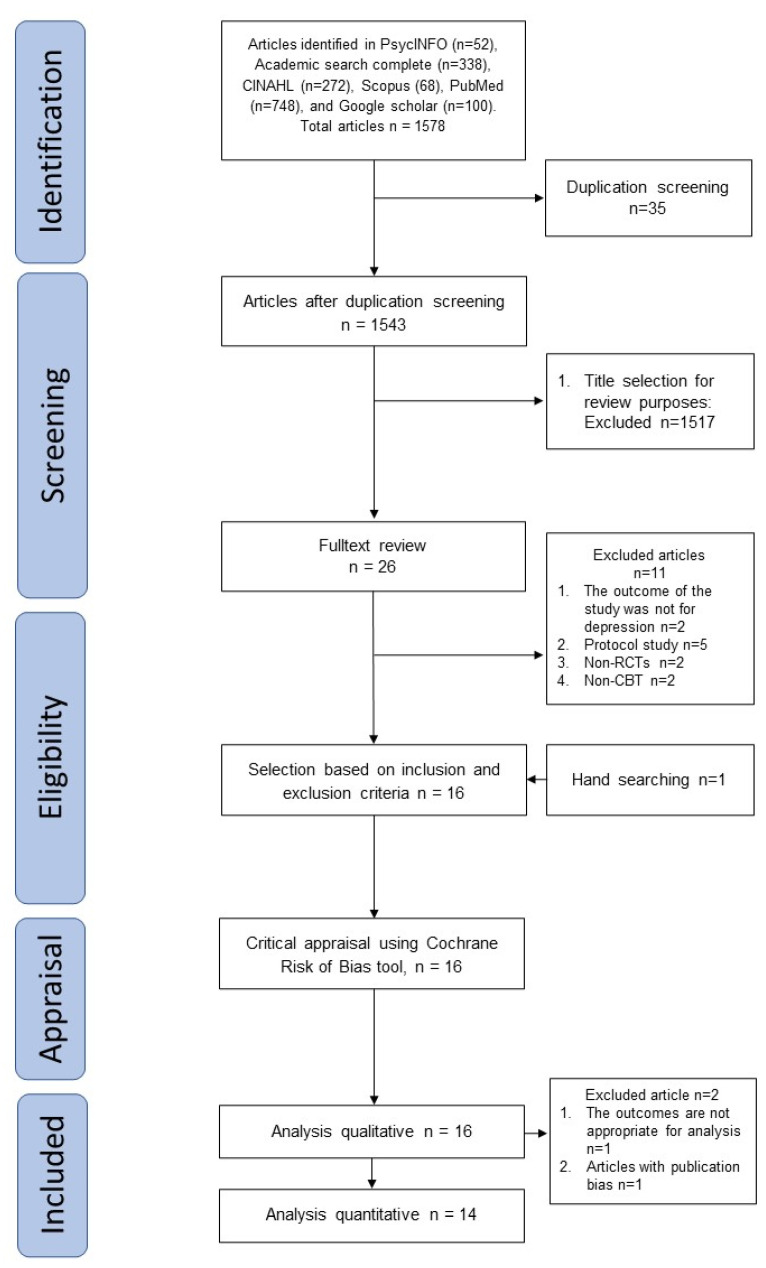
PRISMA flow diagram.

**Figure 2 healthcare-11-00943-f002:**
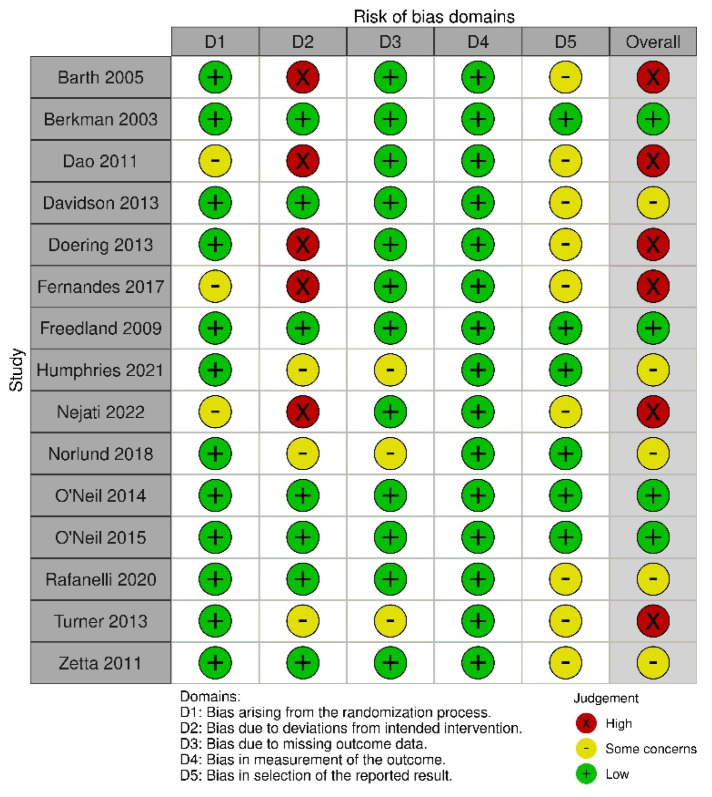
Risk-of-bias assessment [29,30,31,32,33,34,35,36,37,38,39,41,42,43,47].

**Figure 3 healthcare-11-00943-f003:**
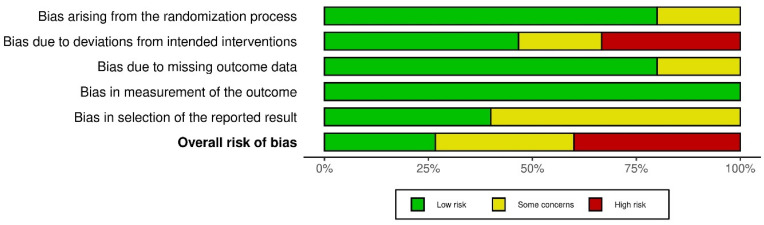
Summary of risk-of-bias assessment.

**Figure 4 healthcare-11-00943-f004:**
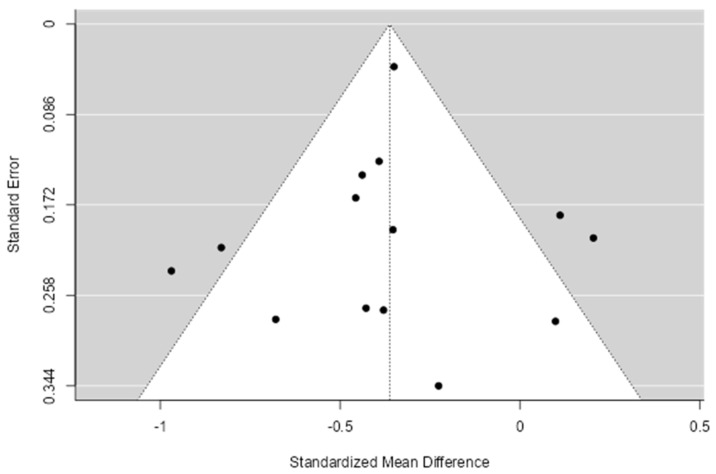
Funnel plot after removing outlier study.

**Figure 5 healthcare-11-00943-f005:**
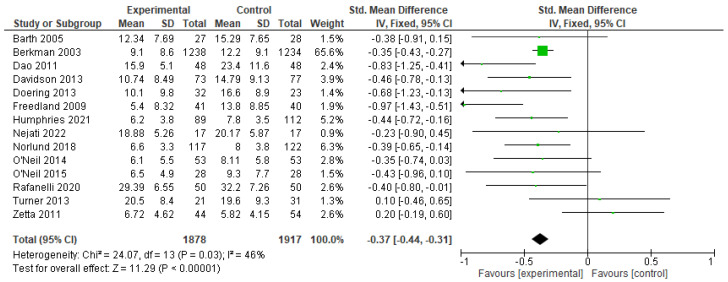
Forest plot describing efficacy of CBT in reducing depression among patients with CHD [29,30,31,32,33,35,36,37,38,39,41,42,43,47]. The arrow symbol in the figure shows the standard mean difference range and the green box shows the standard mean difference in each study.

**Table 1 healthcare-11-00943-t001:** Publication bias assessment accounting for the outlier study.

Test Name	Value	*p*
Fail-safe N	377.000	<0.001
Begg and Mazumdar rank correlation	0.011	1.000
Egger’s regression	−0.013	0.989
Trim and fill number of studies	0.000	-

Note: Fail-safe N was calculated using the Rosenthal approach.

## Data Availability

Not applicable.

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
