# Peer review of "Efficacy of Cognitive Behavior Therapy in Reducing Depression among Patients with Coronary Heart Disease: An Updated Systematic Review and Meta-Analysis of RCTs"

_healthcare, 2023, doi:10.3390/healthcare11070943_

Round 1
Reviewer 1 Report
The reviewed work is a meta-analysis of research on the effectiveness of cognitive behavioral therapy (CBT ) in the treatment of depression accompanying coronary heart disease (CHD). The paper is written correctly. However, in the discussion of research the lack of effectiveness of psychodynamic therapy/lack of research on this topic should be highlighted for contrast.
Author Response
Dear The First Reviewer of MDPI Healthcare,
Thank you for allowing me to submit a revised draft of our manuscript titled “Efficacy of Cognitive Behavior Therapy in Reducing Depression Among Patients with Coronary Heart Disease: An Update of a Systematic Review and Meta-Analysis of RCTs”. We appreciate the time and effort you have dedicated to providing your valuable feedback on our manuscript. We are grateful to you for your insightful comments on our paper. We have been able to incorporate changes to reflect most of the suggestions provided by the reviewers. We have highlighted the changes within the manuscript. Here is a point-by-point response to your comments and concerns. We hope that the manuscript can published in the MDPI Healthcare.
The reviewed work is a meta-analysis of research on the effectiveness of cognitive behavioral therapy (CBT) in the treatment of depression accompanying coronary heart disease (CHD). The paper is written correctly. However, in the discussion of research the lack of effectiveness of psychodynamic therapy/lack of research on this topic should be highlighted for contrast.
Response:
Thank you for your corrections. We have added the lack of therapy’s effectiveness in the discusssion section in lines 498-513, as below:
“In addition to the effectiveness of CBT and its characteristics in reducing depression among CHD patients, this study also recognized several things that need concern since they can be at risk of reducing the efficacy of therapy. This decrease in efficacy was identified from two studies that revealed unsatisfactory results [33], [41]. The characteristics of this study were interventions with a short duration of 6 sessions/weeks [41] and 12 sessions but with a short total intervention time, of only 2.5 hours [33]. In addition, the approach in one of the two studies used group intervention [41]. These two characteristics, including short intervention and group approach, were shown to have no significant effect on reducing depression in this meta-analysis. This result is also reinforced by the research of Christensen et al. who concluded that extended CBT was more effective in reducing depression than brief CBT [63]. Likewise, with a group and individual therapy, some studies show that individuals were more effective [64]. However, some indicated effectiveness that was just as effective [65]. So that these two things are thought to influence the efficacy of CBT in reducing depression, although it is possible that other factors that have not been identified influence these results. However, it is better to cautiously consider brief intervention and group approach in CBT to reduce depression in this patient population.”
Reviewer 2 Report
Congratulations for your work, I am enclosing some clarifications that I believe could enrich the review:
- The introduction is somewhat abbreviated; it should be expanded to include more literature references.
- There is no indication of the date when each search resource was last consulted.
- There is a brief and very general description that 1517 articles were discarded because their title was not relevant; in view of the enormous number of articles discarded, it would be useful to have a more detailed description of the criteria for non-selection.
- The methods for the assessment of the level of certainty of the evidence should have been specified.
- The study reports limitations of evidence included in the study, but not limitations of the review process.
- It is not explicit whether the review is registered; if so, the registration number should be part of the description.
Author Response
Dear The Second Reviewer of MDPI Healthcare,
Thank you for allowing me to submit a revised draft of our manuscript titled “Efficacy of Cognitive Behavior Therapy in Reducing Depression Among Patients with Coronary Heart Disease: An Update of a Systematic Review and Meta-Analysis of RCTs”. We appreciate the time and effort you have dedicated to providing your valuable feedback on our manuscript. We are grateful to you for your insightful comments on our paper. We have been able to incorporate changes to reflect most of the suggestions provided by the reviewers. We have highlighted the changes within the manuscript. Here is a point-by-point response to your comments and concerns. We hope that the manuscript can published in the MDPI Healthcare.
- The introduction is somewhat abbreviated; it should be expanded to include more literature references.
Response:
Thank you for your corrections. We have revised the paragraph in the introduction section, which already added 8 additional pieces of literature to increase references (In lines 49 to 53), They are:
- “Treatment of depression in CHD patients classified in various studies generally consists of psychosocial therapy [14]–[16], pharmacological therapy [17], exercises [18], [19], nutrition [20], and spiritual therapy [21], [22]. Of these many therapies, psychosocial therapy is the most frequently given therapy to reduce depression in patients with cardiac problems and experiencing depression.”
- There is no indication of the date when each search resource was last consulted.
Response:
Thank you for your corrections. We have revised and added the last search consulted in each resource in the search strategy section (lines 126-134). They are:
- We performed a structured literature search independently by two authors (A.N, S.S.) using five databases: CINAHL (accessed on October 30, 2022), PubMed (accessed on October 30, 2022), Scopus (accessed on October 30, 2022), Academic Search Complete (accessed on October 30, 2022), and PsycInfo (accessed on October 31, 2022), also search engine Google Scholar (accessed on October 31, 2022).
- There is a brief and very general description that 1517 articles were discarded because their title was not relevant; in view of the enormous number of articles discarded, it would be useful to have a more detailed description of the criteria for non-selection.
Response:
Thank you for your corrections. We have added the description of criteria for non-selection title articles in the study selection section (Lines 203-206). As below:
- “We excluded 35 duplicate articles and another 1517 studies because the article's title was irrelevant. The title was irrelevant because it indicated that it was not an intervention study, the respondents were non-cardiac patients, or the research method was a review.”
- The methods for the assessment of the level of certainty of the evidence should have been specified.
Response:
Thank you for your corrections. We have added the description of the assessment of the level of certainty of the evidence in table 2 in the appraisal results (supplementary files) and described in the study selection section (lines 139-145), as below:
- Finally, we checked each against the Joanna Briggs Institute (JBI) critical appraisal check-list for RCTs. Specifically, we calculated the critical appraisal scores as the number of “yes” responses divided by the total number of “unclear,” “no,” and “yes” responses, excluding any “no information” responses [34]. Following our appraisals, we eliminated any study with a score < 70%. Furthermore, the third author (Y.T) provides a decision if there was a discrepancy in the selection results. We experienced no disagreements regarding paper eligibility. The critical appraisal score is shown in table 1 in the supplementary files.
- The study reports limitations of evidence included in the study, but not limitations of the review process.
Response:
Thank you for your corrections, We have added the weaknesses of the included studies, without eliminating the weaknesses in the data from the analysis carried out, because this can be input for updating unanswered information from this research (Lines 523-529). As below:
- “Meanwhile, the weaknesses in included studies might arise from low internal validity in some aspects. Some participants were aware of the intervention, which can increase the risk of outcomes assessment bias on the results. In addition, several studies showed low participant compliance with interventions, especially in internet-based CBT, which can raise the risk of increased deviation from the intended intervention. Future research needs to improve these weaknesses by applying participant blinding and increasing participant compliance.”
- It is not explicit whether the review is registered; if so, the registration number should be part of the description.
Response:
Thank you for your suggestions. Protocol registration of systematic reviews/meta-analysis (SR/MA) is still not mandatory although we realize that it is recommended. We realized this only after the review process was almost complete, while the rules for registering protocols in protocol databases such as PROSPERO state that a review cannot be registered for a protocol, if the review process has reached the analysis stage. In other words, we were late in registering the study protocol. So we add information related to this in the study design, lines 78 – 79, as below:
2.1 Study Design
This study was a systematic review conducted following the Preferred Reporting Items for Systematic Reviews and Meta-analyses guidelines [17]. The study protocol was not published or registered.
Reviewer 3 Report
This research is very valuable.
What was not convincing is that the research cited was not new. Is there a criterion for selecting papers in recent years?
The purpose of the study is to update past research. So it is expected that more recent research will be used. Maybe a lot of time has passed between the time this research was done and the submission of the manuscript, or maybe it is better to state the reason in the limitations section.
Author Response
Dear The Third Reviewer of MDPI Healthcare,
Thank you for allowing me to submit a revised draft of our manuscript titled “Efficacy of Cognitive Behavior Therapy in Reducing Depression Among Patients with Coronary Heart Disease: An Update of a Systematic Review and Meta-Analysis of RCTs”. We appreciate the time and effort you have dedicated to providing your valuable feedback on our manuscript. We are grateful to you for your insightful comments on our paper. We have been able to incorporate changes to reflect most of the suggestions provided by the reviewers. We have highlighted the changes within the manuscript. Here is a point-by-point response to your comments and concerns. We hope that the manuscript can published in the Journal of Multidisciplinary Healthcare.
Response:
Thank you for the suggestion, we hope this study can become part of MDPI Healthcare.
What was not convincing is that the research cited was not new. Is there a criterion for selecting papers in recent years?
- The purpose of the study is to update past research. So, it is expected that more recent research will be used. Maybe a lot of time has passed between the time this research was done and the submission of the manuscript, or maybe it is better to state the reason in the limitations section.
Response:
Thank you for your suggestion, based on our knowledge a meta-analysis was carried out to obtain pooled data from studies that have been conducted, especially to improve the external validity aspect, in this case, the generalizability aspect of the effectiveness of an intervention on populations from different places. So that all related research, from the beginning to the most recent should be analyzed and counted so that better-pooled data is obtained, and increases generalizability. Therefore, we continue to recalculate the value of pooled data from previous studies coupled with the latest studies, so that the generalizability aspect becomes wider, and answers the analysis of sub-groups that have not been identified in previous studies such as internet-based CBT. In this meta-analysis, the information related to this question has been answered even though similar studies still need to be expanded in different places. However, these additional studies increased the external validity of the generalizability aspect since the latest studies increased the number of participants from different populations and countries.
The identified updates from this research have been mentioned at the beginning of the discussion, as below:
“The primary purpose of this review was to identify and update the efficacy of CBT in improving depression, and the secondary outcome was to analyze the characteristics of CBT that can effectively improve depression in patients with CHD. There were 14 studies included in the quantitative analysis in this study. In contrast to previous research, 10 of 12 studies discussed CBT's effects on depression in patients with CHD. Updates from this study were identified from four additional studies [20], [28], [29], [33]. The update of this research mainly comes from the four additional recent studies identified from publications in 2015 to 2022, where the former meta-analysis analyzed studies published from 2003 to 2014." (In lines 392-400).
Reviewer 4 Report
healthcare-2202733
The reviewed study aims to perform an update of a systematic review and meta-analysis of the efficacy of CBT in depression among CHD patients. As the authors stated, the manuscript is an update of previously published data, but in my opinion, the scope of the paper and studies included in the analyses justified it (the update provides more interesting data on the topic). However, some of the manuscript areas demand some rework before possible publication. My suggestions are related to:
- Every abbreviation used for the first time should have a full description. Unfortunately, it is not the case for e.g. STEMI, NSTEMI, PCI, AMI, CARG (e.g. this is explained only below the table) and many more also in table 1 (e.g. PST).
- There is no information about the study protocol.
- Because this is an update, authors could provide some description of e.g. how many new studies are now included compared to previous systematic reviews.
- Line 90 – DSM – which one?
- Types of comparators – authors provided a list of a few comparators, but in the end, in table 1, most studies have the usual care control. Maybe data would be more homogenous if only one type of comparator were used. Moreover, what is the difference between standard care and usual care?
- The search strategy could also be presented as a supplement in a more detailed form.
- I would also suggest moving table 1 to the supplementary file.
- Figure 1 – PRISMA has too small font. Moreover, the figure (as well as the description) lacks information about full-text screening.
- Line 166-167 – „Moreover, if studies conveyed multiple depression scores from different scales, the most common scale used by other eligible studies was elected” – which was? Provide that info in e.g. table 1
- As I mentioned earlier, every abbreviation used in table 1 should be explained below the table. It also includes CBT (in a few places, there is iCBT)
- The manuscript also needs linguistic correction, especially in the case of incorrect use of tenses.
- CRUCIAL – every text description of the result should be presented with a figure number!
- Please provide a figure also for sensitivity analysis and funnel plot (it could be in a supplementary file)
- Sensitivity analysis based on low vs high risk of bias publication is an interesting approach, but this should (in my opinion) be based on heterogeneity. If the analysis presented in the forest plot showed high heterogeneity >60% (even >50%), sensitivity analysis could be calculated for the corresponding result.
- CRUCIAL – Using random or fixed effect models presented in figures should be reworked because using one of them should be related to heterogeneity. If it is >50% authors should use a random model, if below 50% fixed effect model. Moreover, if a random model is applied, inverse variance is a proper statistical method, but when a fixed model is used, Maentel-Henschel should be used. Because of that, a few figures should be redone.
After the proposed adjustment, manuscript clearance, readability, and scientific values should increase.
Author Response
Dear The Fourth Reviewer of MDPI Healthcare,
Thank you for allowing me to submit a revised draft of our manuscript titled “Efficacy of Cognitive Behavior Therapy in Reducing Depression Among Patients with Coronary Heart Disease: An Update of a Systematic Review and Meta-Analysis of RCTs”. We appreciate the time and effort you have dedicated to providing your valuable feedback on our manuscript. We are grateful to you for your insightful comments on our paper. We have been able to incorporate changes to reflect most of the suggestions provided by the reviewers. We have highlighted the changes within the manuscript. Here is a point-by-point response to your comments and concerns. We hope that the manuscript can published in the MDPI Healthcare.
The reviewed study aims to perform an update of a systematic review and meta-analysis of the efficacy of CBT in depression among CHD patients. As the authors stated, the manuscript is an update of previously published data, but in my opinion, the scope of the paper and studies included in the analyses justified it (the update provides more interesting data on the topic). However, some of the manuscript areas demand some rework before possible publication. My suggestions are related to:
- Every abbreviation used for the first time should have a full description. Unfortunately, it is not the case for e.g. STEMI, NSTEMI, PCI, AMI, CARG (e.g. this is explained only below the table) and many more also in table 1 (e.g. PST).
Response:
Thank you for your corrections. We have revised the abbreviation in the section of eligibility criteria (in lines 82-88), types of participants (in lines 91-96), and types of interventions (in lines 98-103). It has also been added in the below of table 2 (supplementary files) which was previously named table 1.
- There is no information about the study protocol.
Response:
Thank you for your corrections. Protocol registration of systematic reviews/meta-analysis (SR/MA) is still not mandatory although we realize that it is recommended. We realized this only after the review process was almost complete, while the rules for registering protocols in protocol databases such as PROSPERO state that a review cannot be registered for a protocol, if the review process has reached the analysis stage. In other words, we were late in registering the study protocol. So we add information related to this in the study design, lines 78 – 79, as below:
2.1 Study Design
This study was a systematic review conducted following the Preferred Reporting Items for Systematic Reviews and Meta-analyses guidelines [17]. The study protocol was not published or registered.
- Because this is an update, authors could provide some description of e.g. how many new studies are now included compared to previous systematic reviews.
Response:
Thank you for your corrections. We have added a description of the paragraph in the discussion (principal results). They are:
“The primary purpose of this review was to identify and update the efficacy of CBT in improving depression, and the secondary outcome was to analyze the characteristics of CBT that can effectively improve depression in patients with CHD. There were 14 studies included in the quantitative analysis in this study. In contrast to previous research, 10 of 12 studies discussed CBT's effects on depression in patients with CHD [46]. Updates from this study were identified from four additional studies [29], [35], [36], [42]. The update of this research mainly comes from the four additional recent studies identified from publi-cations in 2015 to 2022, where the former meta-analysis analyzed studies published from 2003 to 2014. These additional studies increased the external validity of the generalizabil-ity aspect since the latest studies increased the number of participants from different populations and countries..“ (Lines 463 – 474)
- Line 90 – DSM – which one?
Response:
Thank you for your suggestion, we have revised the sentence, and deleted DSM, as below:
The patient has a depression score or has identified depressive symptoms based on a reliable and valid depression measurement tool. (In lines: 95 – 96).
- Types of comparators – authors provided a list of a few comparators, but in the end, in table 1, most studies have the usual care control. Maybe data would be more homogenous if only one type of comparator were used. Moreover, what is the difference between standard care and usual care?
Response:
Thank you for your corrections, revisions related to this type of comparator have been described in lines 109-114 and lines 274 – 283, most of the comparators used are usual care, but waitlist interventions, brief interventions, and clinical management have also been identified. It seems that the last three comparators cannot be included in usual care because they are different in the interventions and the timing of the intervention. It is stated as below:
“Based on identification from the studies analyzed, there are four types of comparators, they were usual care [19]–[30], waitlist intervention [31], brief intervention [32], and clinical management [33]. Most of the identified comparators mentioned in the studies were usual care, where the intervention is in accordance with the treatment program received by the patient at the hospital or cardiac rehabilitation program, such as therapy to reduce risk factors and promote a healthy lifestyle. Another comparator is the provision of brief interventions in the form of giving feedback from the baseline assessment, where the control group did not receive the following CBT treatment [32].” (lines 274 – 283)
- The search strategy could also be presented as a supplement in a more detailed form.
Response:
Thank you for your suggestion, we have added the search strategy in the supplementary file (Table 3).
- I would also suggest moving table 1 to the supplementary file.
Response:
Thank you for your suggestion, we have added table 1 to the supplementary file and renamed it to table 2.
- Figure 1 – PRISMA has too small font. Moreover, the figure (as well as the description) lacks information about full-text screening.
Response:
Thank you for your suggestion, we have revised the PRISMA diagram and rewrite in the study selection (In lines 136-145.). There are:
2.3.2 Study Selection
We then checked the titles, abstracts and fulltext for the relevance regarding the study topic and the inclusion and exclusion criteria. Finally, we checked each fulltext against the Joanna Briggs Institute (JBI) critical appraisal checklist for RCTs. Specifically, we calculated the critical appraisal scores as the number of “yes” responses divided by the total number of “unclear,” “no,” and “yes” responses, excluding any “no information” responses [43]. Following our appraisals, we eliminated any study with a score < 70%. Furthermore, the third author (Y.T) provides a decision if there was a discrepancy in the selection results. Notably, we experienced no disagreements regarding paper eligibility.
- Line 166-167 – „Moreover, if studies conveyed multiple depression scores from different scales, the most common scale used by other eligible studies was elected” – which was? Provide that info in e.g. table 1
Response:
Thank you for your suggestion, we have revised in table 2 in the outcome section (supplementary file). The outcomes have been revised, the data written down is taken from the most common scale, such as in Barth's (2005) study written down taken from BDI II, O'Neil (2014 and 2015) taken from PHQ9. The two instruments are the most common instruments used to identify depressive symptoms in respondents. (Table 1).
- As I mentioned earlier, every abbreviation used in table 1 should be explained below the table. It also includes CBT (in a few places, there is iCBT)
Response:
Thank you for your corrections. We have revised the abbreviation in the section on eligibility criteria (in lines 82-88), types of participants (in lines 90-96), and types of interventions (in lines 98-103). And also added them in the below of table 2 (Supplementary files).
- The manuscript also needs linguistic correction, especially in the case of incorrect use of tenses.
Response:
Thank you for your corrections, we will do proofreading after this manuscript is accepted.
- CRUCIAL – every text description of the result should be presented with a figure number!
Response:
Thank you for your suggestion, we have revised the Sub-Group Analysis sections. We moved the figures from the Sub-Group Analysis results (Appendix) to the main manuscript in the Sub-Group Analysis sections. We have corrected each image according to the suggestions by adding figure captions below the table and adding figure number descriptions to each description in the Sub-Group Analysis sections.
- Please provide a figure also for sensitivity analysis and funnel plot (it could be in a supplementary file)
Response:
Thank you for your suggestion, we have added the figure also for sensitivity analysis and funnel plot in the supplementary files.
- Sensitivity analysis based on low vs high risk of bias publication is an interesting approach, but this should (in my opinion) be based on heterogeneity. If the analysis presented in the forest plot showed high heterogeneity >60% (even >50%), sensitivity analysis could be calculated for the corresponding result.
Response:
Thank you for your suggestion, however, the heterogeneity of studies was less than 50%, so we chose to analyze the sensitivity using a fixed effect model. The results are shown in figures 26 and 27 in supplementary files.
- CRUCIAL – Using random or fixed effect models presented in figures should be reworked because using one of them should be related to heterogeneity. If it is >50% authors should use a random model, if below 50% fixed effect model.
Response:
Thank you for your suggestion, we have added information regarding this in the data synthesis and we have re-checked and improved the data analysis based on the amount of heterogeneity, especially in the subgroup analysis section (the Forest plot has been re-worked). As below:
“In addition, if the I2 value less than 50%, the analysis used the fixed effect model, and if I2 value was more than 50%, the analysis used a random effect mode.” (In lines 174-176)
- Moreover, if a random model is applied, inverse variance is a proper statistical method, but when a fixed model is used, Maentel-Henschel should be used. Because of that, a few figures should be redone.
Response:
Thank you for your suggestion, correct us if we have a misunderstanding, based on the literature that I studied said that:
“ RevMan implements two random-effects methods for dichotomous data: a Mantel-Haenszel method and an inverse-variance method. The difference between the two is subtle: the former estimates the amount of between-study variation by comparing each study’s result with a Mantel-Haenszel fixed-effect meta-analysis result, whereas the latter estimates the amount of variation across studies by comparing each study’s result with an inverse-variance fixed-effect meta-analysis result. In practice, the difference is likely to be trivial. The inverse-variance method was added in RevMan version 5.”
Source:
https://handbook-5-1.cochrane.org/chapter_9/9_4_4_3_random_effects_method.htm
Based on this reference, inverse-variance can be used for the fixed-effect model. In addition, based on other references in the JBI reviewers manual (2020) page 95, it states:
“Different statistical methods are available for meta-analysis: Mantel-Haenszel method, Peto’s method, DerSimonian and Laird method, and the inverse variance method. The Mantel-Haenszel method, the Peto’s method, and the inverse variance method are methods used with the fixed effects model of metaanalysis (Deeks et al 2008). The DerSimonian and Laird method is used with the random effects model of meta-analysis (Deeks et al 2008). The inverse variance method may be used with all types of ratios and differences for example the log odds ratio, log relative risk, risk difference, mean difference (weighted mean difference) and standardized mean difference (Petitti 2000; Deeks et al 2008). The Mantel–Haenszel method may be used with ratios, typically with odds ratio, but can be applied to rate ratio and risk ratio (Petitti 2000).”
(Source: Joanna Briggs Institute. (2020). JBI Reviewer ’ s Manual. In The Joanna Briggs Institute (Issue March). https://reviewersmanual.joannabriggs.org/).
So that, based on these references, we determined that the most appropriate statistical method should use inverse variance methods because this research uses the standardized mean difference to calculate pooled data. Furthermore, when I enter the data with type “continues”, with the analysis of “the fixed effect model” and “effect measure std. mean difference”, we cannot select the Mantel-Haenszel method in the default review manager 5.4.1 mode. In the default menu the inverse variance has been selected, as shown in the figure below:

Round 2
Reviewer 3 Report
Corrections to the manuscript improved it. It is acceptable in its present form
Author Response
Dear Reviewer,
Thank you for the suggestions that have been given to us so that we can improve the quality of our manuscripts. We appreciate the time and effort you have dedicated to providing your valuable feedback on our manuscript. We are grateful to you for your insightful comments on our paper. Thank You
Sincerely,
Authors